

# LTA4H extensively associates with mRNAs and lncRNAs indicative of its novel regulatory targets

Tianjiao Ren, Song Wang, Bo Zhang, Wei Zhou, Cansi Wang, Xiaorui Zhao and Juan Feng

Department of Otorhinolaryngology, The First Affiliated Hospital of Xinjiang Medical University, Urumqi, Xinjiang, China

## ABSTRACT

The RNA-binding metabolic enzyme LTA4H is a novel target for cancer chemoprevention and chemotherapy. Recent research shows that the increased expression of LTA4H in laryngeal squamous cell carcinoma (LSCC) promotes tumor proliferation, migration, and metastasis. However, its mechanism remains unclear. To investigate the potential role of LTA4H in LSCC, we employed the improved RNA immunoprecipitation and sequencing (iRIP-Seq) experiment to get the expression profile of LTA4H binding RNA in HeLa model cells, a cancer model cell that is frequently used in molecular mechanism research. We found that LTA4H extensively binds with mRNAs/pre-mRNAs and lncRNAs. In the LTA4H binding peak, the frequency of the AAGG motif reported to interact with TRA2$\beta$4 was high in both replicates. More notably, LTA4H-binding genes were significantly enriched in the mitotic cell cycle, DNA repair, RNA splicing-related pathways, and RNA metabolism pathways, which means that LTA4H has tumor-related alternative splicing regulatory functions. QRT-PCR validation confirmed that LTA4H specifically binds to mRNAs of carcinogenesis-associated genes, including LTBP3, ROR2, EGFR, HSP90B1, and lncRNAs represented by NEAT1. These results suggest that LTA4H may combine with genes associated with LSCC as an RNA-binding protein to perform a cancer regulatory function. Our study further sheds light on the molecular mechanism of LTA4H as a clinical therapy target for LSCC.

## INTRODUCTION

Laryngeal cancer is not only the most common malignancy of the head and neck, but also the second most common respiratory system tumor after lung cancer. 95 to 98% of laryngeal cancers are squamous cell carcinomas (LSCC) (*Zuo et al., 2016*). Research shows that nearly 180,000 new cases of throat cancer and nearly 100,000 throat cancer deaths were reported worldwide in one year (*Bray et al., 2018*). Due to the proneness to recurrence and metastasis, the five-year overall survival rate for LSCC has been approximately 50% in recent years (*Cavaliere et al., 2021*). Due to LSCC's uncertain molecular mechanism, over 60% of patients were already at late stages when the disease was discovered (*Steuer et al., 2017*). An increasing amount of research demonstrates that the formation and development

Corresponding author
Juan Feng, 19773913@qq.com

of LSCC is related to molecular mechanisms. Studies have shown that the prognosis of patients with LSCC with down-regulated HLA class I antigen was worse (*Ogino et al., 2006*). It has also been reported that the ANXA1 interaction with FPR2/ALX promote proliferation and metastasis of LSCC (*Gastardelo et al., 2014*). Studies have revealed lincRNA HOTAIR is highly expressed in LSCC and promotes methylation of PTEN (*Li et al., 2013*). Hence, to better identify biomarkers and explore effective new therapeutic strategies, it is essential to reveal the LSCC's carcinogenic mechanism.

Genes, miRNAs and lncRNAs all play key roles in tumor genesis and development. In recent years, researchers have found that many genes, miRNAs and lncRNAs are key factors of LSCC (*Zhang et al., 2016*). *Gao et al. (2019)* detected LTA4H expression in LSCC and normal tissue by qPCR. The results showed that LTA4H was significantly up-regulated in LSCC tissues than in normal tissue. A recent study screened 275 differential proteins associated with laryngeal cancer through PPI (protein-interaction) network analysis. GO function was significantly enriched in RNA processing and respiratory electron transport chains, and LTA4H was one of the up-regulated proteins (*Peyvandi et al., 2018*). More importantly, studies have demonstrated that increased LTA4H expression in LSCC is associated with a poor prognosis, and knockdown of LTA4H successfully suppresses the growth, invasion and migration of laryngeal carcinoma cells (*Gao et al., 2019*; *Peyvandi et al., 2018*). However, more study is necessary to fully understand the specific molecular mechanism of LTA4H in laryngeal carcinoma.

There is increasing evidence that LTA4H is overexpressed in many malignant cancers, which promotes cancer cell proliferation. For example, it has been identified that LTA4H is overexpressed in esophageal adenocarcinoma and through inflammation-augmenting effect and growth-stimulatory effect to promote carcinogenesis (*Chen et al., 2004*). Studies have also shown that LTA4H can enhance aminopeptidase and epoxide hydrolase activity to promote colon cancer growth (*Jeong et al., 2009*). Activation of 5-LOx/LTA4H can stimulate oral epithelial cell proliferation and inflammation, which is the main way to promote oral cancer (*Guo et al., 2011*; *Sun et al., 2006*).

As a zinc-dependent epoxide hydrolase and aminopeptidase, the active site of Leukotriene A4 hydrolase (LTA4H) can be the target action site of related inhibitors (*Chen et al., 2004*; *Haeggström, 2004*; *Vo, Jang & Jeong, 2018*). LTA4H is being investigated as a new target for cancer treatment due to its role in inflammatory response and tumor progression. As a hydrolase, LTA4H performs its classic biological functions, including chemotaxis, endothelial cell adhesion, and leukocyte activation, by acting on the last step of the arachidonic acid metabolic process (*Haeggström, 2018*; *Oh & Olefsky, 2016*; *Snelgrove et al., 2010*). As an aminopeptidase, it involves in inflammation and host defensed though grading proline-glycine-proline (PGP), a neutrophil chemokine that is also a biomarker for chronic obstructive pulmonary disease (*Haeggström, 2004*; *Snelgrove et al., 2010*). Furthermore, LTA4H is believed to function as an RBP in the post-transcriptional control of specific mRNAs (*Castello et al., 2012b*; *Castello, Hentze & Preiss, 2015*). RBPs not only interact with mRNAs directly, but also bound to proteins and other diverse RNAs to play crucial roles (*Gerstberger, Hafner & Tuschl, 2014*; *Hamosh et al., 2005*). A growing body of evidence show RBPs can promote cancer cell growth, angiogenesis, and metastasis by

regulating numerous target genes related to tumor development (*Kang, Lee & Lee, 2020*). We hypothesized that LTA4H might interact with the RNAs of cancer-related critical genes at the transcriptional or post-transcriptional levels to control the expression of those genes, thus affecting the proliferation, invasion and metastasis of tumors (including LCSS) cells. However, whether LTA4H binds to mRNAs in cancer cells remains unclear. We hypothesized that LTA4H might interact with the RNAs of cancer-related critical genes at the transcriptional or post-transcriptional levels to control the expression of those genes.

To validate our hypotheses, we used improved RNA immunoprecipitation and sequencing (iRIP-Seq) method (*Ke et al., 2021*) on LTA4H in modal HeLa cells (*Capouillez et al., 2009*) to explore its RNA-binding characteristics in cancer cells. The finding demonstrates that LTA4H extensively binds to mRNAs/ pre-mRNAs and lncRNAs. And we have identified some crucial LTA4H-bound genes that regulate cancer development, like NEAT1, LINC00657, LTBP3 and ROR2. These results reveal the underlying molecular mechanisms of LTA4H as a clinical therapeutic target for LCSS, which has significant effects on diagnostic and therapeutic applications.

## MATERIALS & METHODS

### Cloning and plasmid construction

Hot fusion primer pairs were created using CE Design V1.04. Each primer contained a 17–30 bp sequence from the pIRES-hrGFP-1a vector and a gene-specific sequence fragment.

F-primer: agcccgggcggatccgaattcATGCCCGAGATAGTGGATACCTG

R-primer: gtcatccttgtagtcctcgagATCCACTTTTAAGTCTTTCCCCAC.

At 37 °C for 2 to 3 h, we digested the pIRES-hrGFP-1a vector with EcoRI and XhoI (NEB). The enzyme-digested vector was purified on a Qiagen column kit using 1.0% agarose gel. HeLa cells' total RNA was obtained using Trizol. Oligo dT primers were used to transcribe the purified RNA for cDNA. Following that, PCR amplification was used to synthesize the inserted fragment. PCR insert and linearized vector digested with EcoRI and XhoI (NEB) were combined in a PCR microtube and ligated with Vazyme's ClonExpress® II One Step Cloning Kit (Vazyme, Nanjing, Jiangsu, China). Chemical transformation was used to introduce plasmids into *Escherichia coli* strains. We incubated cells overnight at 37 °C on LB agar plates containing 1uL/ml ampicillin. Finally, 28 cycles of colony PCR were performed on the backbone vectors using universal primers to screen colonies.

### Cell culture and transfections

The China Center for Type Culture Collection (CCTCC), Wuhan, Hubei, China provided human cervical carcinoma (CC) cell lines, HeLa (CCTCC@GDC0009). In Dulbecco's modified Eagle's medium (DMEM), which contains 10% fetal bovine serum (FBS), 100 ug/mL streptomycin and 100 U/mL penicillin, we cultivated HeLa cells at 37 °C and 5% $CO_2$. Following the manufacturer's instructions, HeLa cells were transfected using Lipofectamine 2000 (Invitrogen, Carlsbad, CA, USA). Transfected cells were collected after 48 h for RT-qPCR and western blot analyses.

## Assessment of gene overexpression

To evaluated the effect of LTA4H overexpression, we used GAPDH (glyceraldehyde 3-phosphate dehydrogenase) as a control gene. The synthesis of cDNA was carried out according to standard procedures, and we performed RT-qPCR using Bestar SYBR Green RT-PCR Master Mix (DBI Bioscience, Shanghai, China) on a Bio-Rad S1000. Additional file contains primer information. After normalizing to GAPDH mRNA concentration level, each transcript was quantified using 2-$\Delta\Delta$CT method (*Livak & Schmittgen, 2001*). The GraphPad Prism software (San Diego, CA, USA) was then used for comparison with the paired Student's *t*-test.

## Immunoprecipitation

HeLa cells were lysed on ice for 5 min with ice-cold lysis buffer (1 ×PBS, 0.5% sodium deoxycholate, 0.1% SDS, 0.5% NP40) containing RNase inhibitors (Takara, 2313) and protease inhibitors (329-98-6; Solarbio). In order to remove cell debris, the mixture was forcefully shaken and centrifuged for 20 min at 13,000 × g at 4 °C. The centrifuged supernatant was incubated overnight at 4 °C with DynaBeads protein A/G bound to normal IgG or anti-Flag LTA4H antibody. The beads were washed twice with low salt washing buffer, high salt washing buffer and 1X PNK buffer solution respectively, and the samples were suspended in the Elution Buffer to extract RNA from the LTA4H-RNA complex.

## Western blot

Resuspend sample with 40 ul Elution Buffer 50 mM Tris-Cl (PH = 8.0), 10mM EDTA (PH = 8.0), 1%SDS; incubate it at 70 °C, 1,400 rpm for 20 min. The supernatant was put in a fresh EP tube. The complex was separated on a 10% SDS-PAGE gel after being boiled in boiling water with 1X SDS sample buffer for 10 min. with TBST buffer (20 mM Tris-buffered saline and 0.1% Tween-20) contained 5% non-fat milk power, we diluted the primary antibody: flag antibody (1:2,000, F7425; Sigma), actin (1:2,000, 66CUSABIO). The membranes were soaked in the primary antibody incubation solution and incubated at room temperature for 1 h. The membranes were then soaked in the HRP-conjugated secondary antibody incubation solution and incubated at room temperature for 1 h. The enhanced chemiluminescence (ECL) reagent (170506; Bio-Rad, Hercules, CA, USA) was used to detect the binding secondary antibody (anti-mouse or anti-rabbit 1:10,000) (Abcam).

## iRIP-seq library preparation and sequencing

TRIzol (Invitrogen) was used to isolate the RBP-bound RNAs from the immunoprecipitation of anti-Flag. In accordance with the manufacturer's instructions, complementary DNA (cDNA) libraries were prepared using KAPA RNA Hyper RNA binding protein connects the future Prep Kit (KK8541; KAPA). On the Illumina HiSeq X Ten platform, the cDNA libraries were sequenced for 150 bp paired-ends.

## Data analysis

Only uniquely mapped reads were used for the subsequent analysis after reads were matched onto the genome using TopHat 2 (*Kim et al., 2013*) ."ABLIRC" strategy was utilized to

determine the genomic locations where LTA4H binds (*Xia et al., 2017*). Peaks were formed from reads that had at least one base pair of overlap. Using computational simulation, reads with the same number and lengths as reads in peaks were generated randomly for each gene. For the purpose of generating random max peak height from overlapping reads, the outputting reads were further mapped to the same genes. The whole procedure was done 500 times. All observed peaks with heights greater than those of random maximum peaks (*p*-value 0.05) were chosen. The simulation independently analyzed the IP and input samples, removing the IP peaks that overlapped the input peaks. The peaks were used for motifs analysis with the Hypergeometric Optimization of Motif Enrichment (HOMER) software (*Heinz et al., 2010*).

### Functional enrichment analysis

GO term and KEGG path enrichment analysis was performed using KOBAS 2.0 server (*Xie et al., 2011*). According to the annotation information of peak associated gene, the GO Term of each gene was counted, and significance of each Term was analyzed by Benjamini–Hochberg FDR (BH) and hypergeometric test to determine the degree of enrichment.

### Reverse transcription qPCR validation

RT-qPCR was performed using total RNA from the iRIP-seq library preparation. Using the M-MLV Reverse Transcriptase (Vazyme), RNA was reverse transcribed into cDNA. Real-time PCR was carried out with the StepOne RealTime PCR System using the HieffTM qPCR SYBR® Green Master Mix (Low Rox Plus; Yeasen, Pudong, China). Denaturation at 95 °C for 5 min was followed by 40 cycles of denaturation at 95 °C for 15 s, annealing and extension at 60 °C for 30 s under PCR cycling conditions. PCR amplifications were carried out in triplicate for each sample.

### Statistical analysis

The statistical software SPSS 16.0 (Chicago, IL, USA) was used to manipulate the experimental data, which were all presented as mean standard deviation (SD). All experiments were run at least three times independently, and $P < 0.05$ was considered significant.

## RESULTS

### Deregulated expression of LTA4H in various cancers

Previous researches have shown that LTA4H is significantly expressed in several malignancies and affects the initiation and growth of tumors (*Chen et al., 2004*; *Guo et al., 2011*; *Jeong et al., 2009*; *Sun et al., 2006*). In order to explore the relationship between LTA4H and laryngeal squamous cell carcinoma (LSCC), or more broadly head and neck squamous cell carcinoma (HNSCC), we first studied the expression level of LTA4H in LSCC tissues and normal tissues through The Cancer Genome Atlas (TCGA) database (Fig. 1A). Box plot and scatter diagram were used to display the expression levels of LTA4H (Transcripts Per Million (TPM)). According to the finding, LSCC tissues had lower levels of

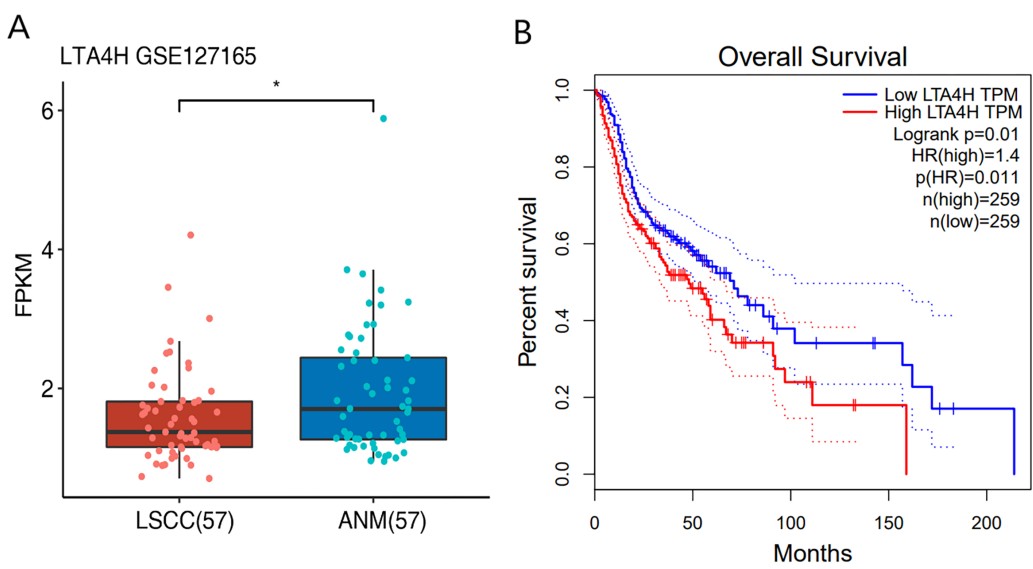

**Figure 1** **Expression and survival analysis of LTA4H.** (A) Transcription levels of LTA4H in LSCC and normal samples from The Cancer Genome Atlas (TCGA) database. (B) Survival analysis of LTA4H in HNSCC from The Cancer Genome Atlas (TCGA) database.

LTA4H than normal control tissues. The results obtained by *Gao et al. (2019)* with TCGA database are also LTA4H downregulation in LSCC. However, the author then detected LTA4H expression in LSCC and normal tissue by qPCR, and have proved that TA4H in LSCC tissues was significantly up-regulated (*Gao et al., 2019*). In addition, we also studied the association between LTA4H expression level and prognosis in HNSCC and normal tissues (Fig. 1B). A blue curve represented the low expression group, while a red curve represented the high expression group. A significant finding was that HNSCC patients with high expression of LTA4H had a poor prognosis. Therefore, the potential function of LTA4H in laryngeal squamous cell carcinoma deserves further study.

## Characterization of the LTA4H-RNA interaction map by iRIP-seq analysis

To explore the potential function of LTA4H in LSCC, we obtained a LTA4H-bound RNA profile in modal HeLa cells by applying theiRIP-seq approach. The iRIP-seq is an advanced technique for studying RBPs, which achieves the precision of CLIP-seq to obtain both direct and indirect binding sites of protein and RNA accurately, whereas maintains the simplicity of RIP-seq. Labelled antibody and control antibody were used for immunoprecipitation, and two separately replicate experiments were performed. For immunoprecipitation, two separate iRIP repetitions were carried out using flag-tagged LTA4H. The western blots of both IP samples showed the presence of the protein Flag- LTA4H, but the IgG control did not (Fig. 2A). Then, we performed paired-end sequencing for the cDNA libraries using the Illumina HiSeq X Ten platform, and obtained the high-quality clean reads. After removing the adapter sequences and low-quality reads, we were left with 30,619,638 and 49,122,624 reads for IP group and input control of replicate 1, and 28,885,522 and 40,002,548 reads from those of the replicate 2 (Table S1). Next, using TopHat 2, we mapped the sequencing
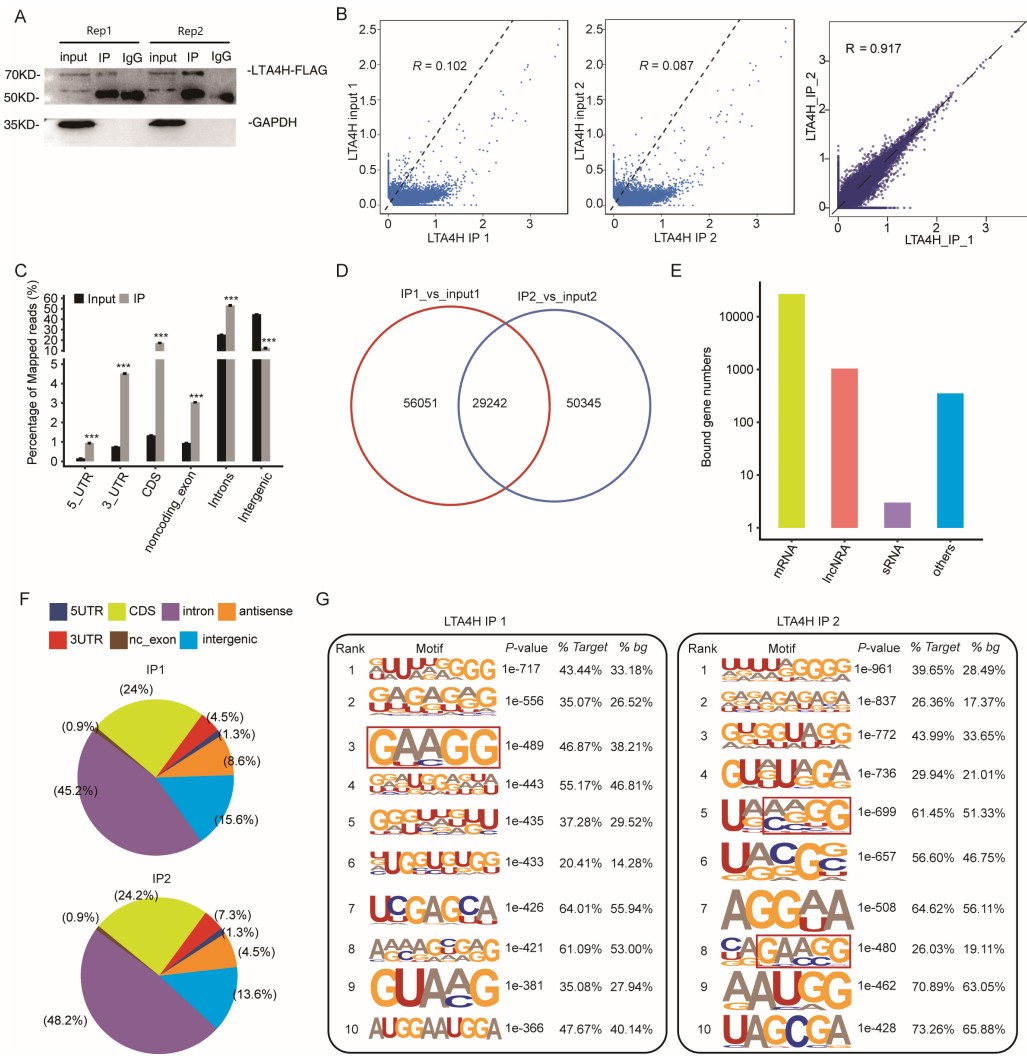

**Figure 2** **Transcriptome-wide identification of LTA4H binding targets using iRIP-seq method.** (A) Western blotting analysis of LTA4H expression. (B) Scatter plot showing Pearson correlation between IP and input samples and between the two IP replicates. (C) Reads distribution across reference genome. Error bars represent mean ± SEM. *** $p < 0.001$, ** $p < 0.01$. (D) Venn diagram showing the overlap of LTA4H binding peaks obtained from two replicates of iRIP-seq. The peaks were called by ABLIRC algorithm. (E) Bar plot showed the classification of LTA4H targets in common in two replicates. (F) Distribution of peaks across reference genome. (G) The top ten over represented motifs in LTA4H binding peaks.

reads to reference genomes GRCh38 (*Kim et al., 2013*). About 77.23–78.94% of them were aligned and about 39.85–83.65% were matched uniquely. The uniquely mapped reads are overwhelmingly from mature mRNAs. The percentage of splice reads among the uniquely aligned reads was substantially greater in the IP sample compared with the control sample, indicating that LTA4H might have an involvement in the splicing role (Table S2).

Correlation analysis of IP and input samples by comparing Reads per kilo base of a gene per million reads (RPKM) (*Mortazavi et al., 2008*) of the same gene revealed transcripts were obviously enriched in IP samples than input control, which indicated that the specificity

of the LTA4H-bound RNA was good (Fig. 2B). We also made correlation analysis between the two IP replicates, and the results showed that $R = 0.917$, which indicated that the two groups of IP samples had good repeatability (Fig. 2B). The results of these two groups of samples are almost the same, which indicated that the iRIP-seq experiment is reliable. The reads distribution across reference genomic regions showed LTA4H binding reads tend to concentrate in the CDS, the intron regions than input control reads, as well as in 3′ UTR, 5′ UTR and noncoding exons regions (Fig. 2C).

In order to eliminate the interference caused by gene expression quantity for predicting LTA4H specific binding sites, we adopted the ABLIRC method (*Chi et al., 2009*) to identify LTA4H-bound peaks precisely. There were 29,242 overlapping peaks in the two replication groups in Hela cells, indicating the overlap of peaks from the two sets of experiments is relatively high (Fig. 2D). Interesting, after sorting according to the number of reads on overlapping peaks, the top peaks were mainly located in mRNAs and lncRNAs (Fig. 2E). The results demonstrated that LTA4H has an extensive capability for RNA binding and may function as a regulator by interacting to mRNAs and lncRNAs. Peak distribution across reference genomic regions revealed that the LTA4H binding peaks located in the intron region accounted for a large proportion (66.93% and 63.20%), followed by CDS region (Fig. 2F).

Then, HOMER was employed to obtain sequence motif enriched within LTA4H peaks. The results showed UG-rich motif and GA-rich motif were presented as the first two motifs of LTA4H peaks of two replicates, which may be the key sites of LTA4H binding to its target (Fig. 2G). We found a high frequency of motif AAGG in both repeats in LTA4H binding peaks. It has recently been reported that TRA2B interacts with motif AAGG to promote cancer cell growth by disrupting gene expression processes associated with aging (*Kajita et al., 2016*). Our results suggest that LTA4H may interact with TRA2B for binding of the motif AAGG to regulate gene expression in cancerous cells.

In conclusion, the obtained LTA4H-binding RNA map will help our understanding of the overall regulatory mechanism of LTA4H-RNA association during gene expression in Hela cells.

## Analysis of pre-mRNA and mRNAs associated by LTA4H

Further, the LTA4H overlapped peak associated genes were compared to the Gene Ontology database for enrichment biological process analysis. We found that LTA4H-bound genes were involved in gene expression, mitotic cell cycle, viral replication and DNA repair (Fig. 3A). Next, the DEseq package was used to identify the LTA4H-bound differentially enriched genes (DEGs) (*Anders & Huber, 2010*). Among the 14,170 DEGs, there were 2,776 enriched genes and 11,394 non-enriched genes related to LTA4H. We constructed a volcanic map to show the significantly enriched genes associated with LTA4H, all of which are associated with oncogenesis, including lncRNAs NEAT1 and LINC00657, and mRNAs ROR2, LTBP3, HSP90B1 and EGFR (Fig. 3B). To explore the potential biological role of these enriched genes, we continued to analyze the enriched genes using the GO database, and it showed they were mainly involved in negative regulation of transcription, mitotic cell cycle, gene expression, and viral replication (Fig. 3C).

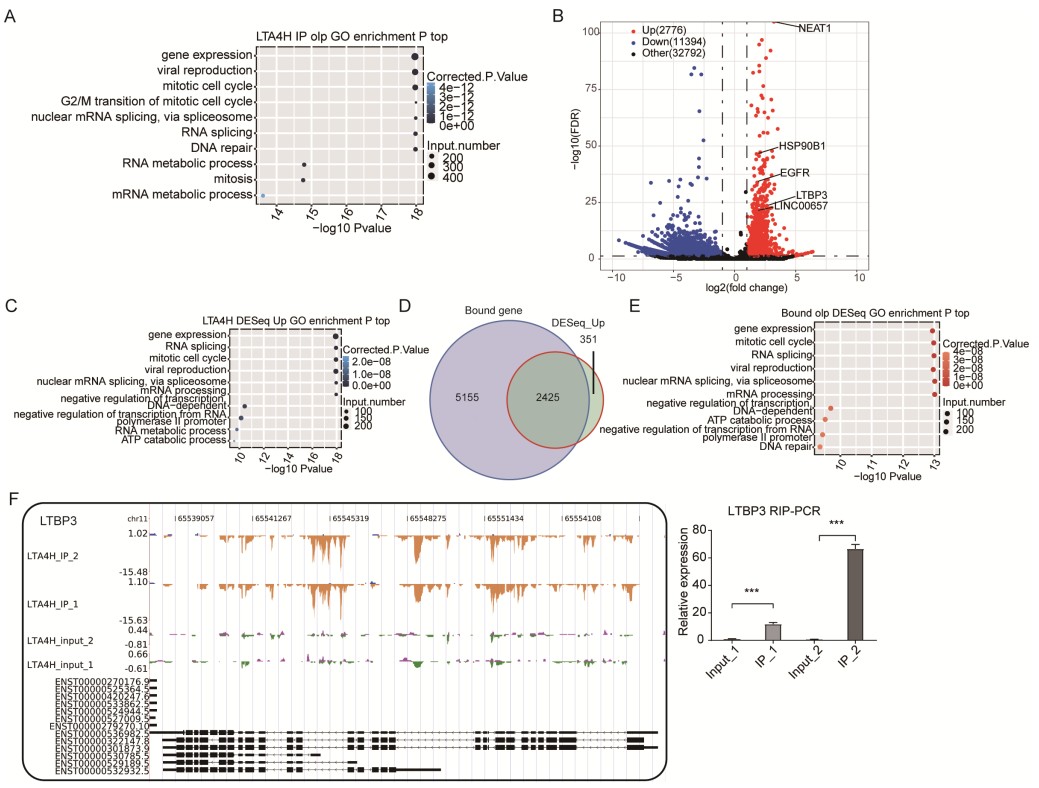

**Figure 3  Analysis of the targets bound by LTA4H.** (A) The top 10 enriched GO biological processes of the LTA4H-bound genes by ABLIRC algorithm in two replicates. (B) Potential targets identified by DEseq. (C) The top 10 enriched GO biological processes of the LTA4H-bound genes by DEseq. (D) Venn diagram showing the overlap of LTA4H bound genes obtained from two replicates by ABLIRC algorithm and DEseq. (E) The top 10 enriched GO biological processes of the LTA4H- bound genes by both ABLIRC algorithm and DEseq. (F) The reads density landscape of LTA4H- binding peaks across LTBP3 (left). Quantification of LTBP3 expression by qRT-PCR using iRIP-seq data (right).

Next, we performed an overlap analysis of LTA4H bound genes from ABLIRC algorithm and DEG from DEseq. Running DEseq identified fewer enriched genes, which were well overlapped by the LTA4H-bound genes by ABLIRC and resulted in 2425 overlapped genes (Fig. 3D). The results demonstrated a significant association between LTA4H-bound and enriched gene expression. GO analysis showed the 2425 overlapped genes were mainly clustered at gene expression, mitotic cell cycle, viral replication and DNA repair (Fig. 3E).

To further verify the presence of Flag-LTA4H protein binding on target genes, we next showed the distribution of reads binding location and coverage depth compared to Peak associated genes. The results across LTBP3 show the two replicates were consistent, and the IP groups were obviously biased towards the intron and exon regions than input control, which was the potential binding region of LTA4H on LTBP3 (Fig. 3F, left panel). And further, we used this gene to verify that it directly interacts with mRNA by RT-qPCR (Fig. 3F, right panel). In comparison to the control group, LTBP3 was considerably higher in the IP group. Similarly, mRNAs EGFR, ROR2 and HSP90B1 were distinctly enriched in IP samples compared to the input samples and the results of subsequent validation were as

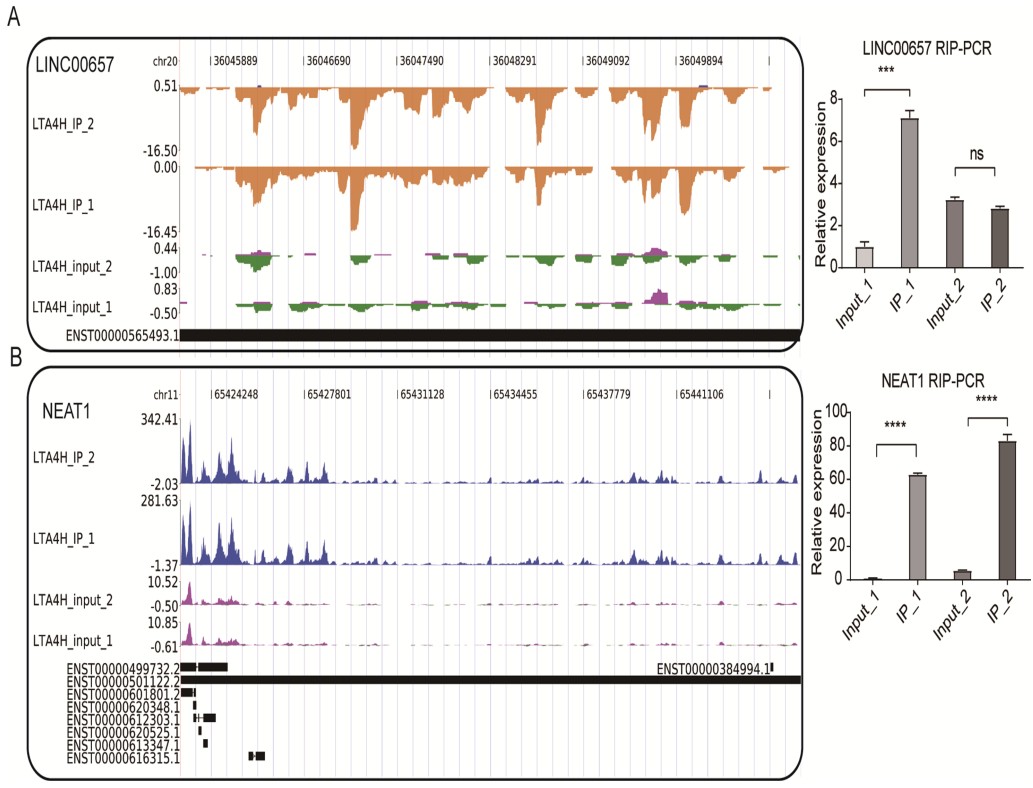

**Figure 4** **LTA4H binds to lncRNA involved in laryngeal squamous cell carcinoma.** (A–B) The reads density landscape of LTA4H-binding peaks across lncRNAs (left). Quantification of LINC00657 and NEAT1 expression by qRT-PCR using iRIP-seq data (right).

expected (Fig. S1). Taken together, our results suggest that LTA4H and mRNAs are closely interacted in Hela cells.

## Analysis of LTA4H-bound lncRNAs

We also conducted the reads density landscape for lncRNAs that LTA4H highly enriched. The results across the cancer-promoting gene LINC00657 show there are many LTA4H-bound peaks in IP compared with input (Fig. 4A, left panel). In the validation experiment, LINC00657 was found to be significantly enriched in IP1, and there was no obvious bias in IP2 compared with the control group, which was considered to be related to experimental error (Fig. 4A, right panel). It may also be that the peak site is not enriched in IP2, that is, LTA4H may not bind LINC00657 specifically, they just bind randomly. The exact mechanism of LTA4H binding to LINC00657 has not been fully clarified and needs further study. NEAT1 had an obvious bias in IP groups, and we know that dysregulation of NEAT1 plays a key carcinogenic role (*Chen et al., 2015*; *Wang et al., 2016*) (Fig. 4B, left panel). Besides that, NEAT1 has been verified to be significantly enriched in the IP group (Fig. 4B, right panel). In summary, we hypothesized that LTA4H is preferentially bound to genes related to tumor formation and progression.

## DISCUSSION

As an essential hydrolase for LTB4 production (*Vo, Jang & Jeong, 2018*), upregulated LTA4H has been found to be linked with various malignancies, such as colon, esophageal, and lung cancer (*Chen et al., 2003a*; *Chen et al., 2004*; *Jeong et al., 2009*). In The Cancer Genome Atlas (TCGA) database, the decrease in LTA4H levels in the LSCC was unexpected. We think it could be related to the small number of laryngeal squamous cell carcinoma TCGA database, leading to inconsistency with the previous experimental results (*Gao et al., 2019*; *Peyvandi et al., 2018*). The low expression of LTA4H in laryngeal cancer tissues was consistent with previous studies (*Gao et al., 2019*; *Rodrigues-Lisoni et al., 2010*). Importantly, further data showed that patients' survival times were considerably shorter when their LTA4H expression was higher, suggesting that LTA4H may have a neoplastic role in HNSCC. On the one hand, numerous investigations have revealed that Leukotriene can control tumor growth by influencing interactions between the stromal cells and tumor epithelial cells, creating the favorable conditions for tumor genesis. So inflammatory mediators can be detected in the tumor microenvironment (*Colotta et al., 2009*; *Wang & Dubois, 2010*). On the other hand, two recent mRNA-interacting protein identification studies reported the activity of LTA4H binding to mRNA (*Castello et al., 2012a*; *Castello, Hentze & Preiss, 2015*). Thus, we speculate that LTA4H not only participates in the regulation of cancer through the inflammatory mediator pathway, but also controls the expression of cancer key genes by interacting with mRNA at the transcriptional or post-transcriptional level. However, to understand the specific mechanism of LTA4H in tumor cells, more study is necessary.

Herein, we used iRIP-seq to identify interactions between LTA4H and RNAs in Hela cells. We analyzed the binding characteristics of LTA4H as RNA binding protein binding to RNAs and found that IP groups were highly enriched comparing with input groups. This indicates that many pre-mRNAs /mRNAs are specifically bound by LTA4H, confirming the function of LTA4H binding RNA. Surprisingly, we found that LTA4H targets were not only enriched in mRNAs, but also in lncRNAs, suggesting that LTA4H was also involved in non-coding processes.

We also analyzed the peaks of LTA4H proteins by using the ABLIRC algorithm from iRIP-seq results. The binding peak of LTA4H was mainly enriched in the Intron region and CDS region, indicating that LTA4H has functional RNA targets. Importantly, GO results revealed that LTA4H-bound proteins were considerably overrepresented in pathways associated with cancer, including mitotic cell cycle, DNA repair, RNA splicing related pathways and RNA metabolism pathways. We know that genomic instability is a common feature of most cancer cells, and DNA damage affects genomic stability (*Bröckelmann, De Jong & Jachimowicz, 2020*; *Negrini, Gorgoulis & Halazonetis, 2010*). In addition, defective DNA repair can lead to a predisposition to cancer (*Chen et al., 2003b*). In eukaryotic cells, RNA splicing is a highly complex fine-tuning step in gene expression, while tumor genes are prone to deactivation mutations at splicing sites (*Rhine et al., 2018*). A study indicated that RBPs influence the development of various cancers by controlling the metabolism of many transcripts, which confirms the relevance of our findings (*Pereira, Billaud & Almeida,*

*2017*). These results gave us a new hint that LTA4H may bind to cancer-related lncRNAs and mRNAs and regulate their expression and splicing levels, which may be a previously unknown molecular regulatory mechanism of LTA4H in cancer.

In our study, we obtained six enriched genes associated with carcinogenesis from 14,170 DEGs. Among them, studies have been shown that mRNAs such as ROR2, LTBP3, HSP90B1, and EGFR have some close links with the occurrence and treatment of LSCC. Upregulated ROR2 and Wnt5a have shown to represent poor tumor stage and lymphatic metastasis in LSCC, suggesting that ROR2 was an independent prognostic factor (*Zhang et al., 2017*). Wnt5a, which interacts with RoR2 physically and functionally, has been demonstrated to be related to the growth of many different cancers (*Asem et al., 2016*; *Oishi et al., 2003*). Likewise, early-stage head and neck neoplasm patients with high levels of LTBP3 have a poor prognosis for survival (*Deryugina et al., 2018*). It has been shown that HSP90B1 is regulated by Mir-99a-3p to participate in the pathogenesis of head-neck cancer (*Okada et al., 2019*), and the highly expressed HSP90B1 represents the poor prognosis of many tumors, including breast cancer and lung cancer (*Lin et al., 2020*; *Liu et al., 2019*). EGFR knockdown suppressed LSCC cell growth, infiltration and migration, and EGFR inhibitors were proved to have anti-laryngeal cancer effects *in vitro* and *in vivo* (*Ren, Wang & Qi, 2021*; *Yang et al., 2020*). These results show that LTA4H interacting mRNAs are involved in regulating cancer proliferation, invasion, and metastasis in LSCC.

We also found LTA4H widely binds to lncRNAs such as NEAT1 and LINC00657, which have recently been investigated for a variety of cellular roles (*Mercer, Dinger & Mattick, 2009*; *Wilusz, Sunwoo & Spector, 2009*). Previous studies have reported that many lncRNAs interact with RBPs to play regulatory functions. For example, MALAT1 binds serine/arginine (SR) proteins to regulate alternative splicing (*Tripathi et al., 2010*). And NEAT1 can regulate transcription, miRNA processing and alternative splicing by binding RBP (*Cooper et al., 2014*; *Imamura et al., 2014*; *Jiang et al., 2017*). We also found some links between these targeted genes and laryngeal cancer and other similar cancers. NEAT1 levels were significantly upregulated in LSCC, which predicted a poor prognosis (*Wang et al., 2016*). Others have shown activation of NEAT1 expression also promoted proliferation, migration, and invasion of Esophageal squamous cell carcinoma (*Chen et al., 2015*). LINC00657 has been shown to play a regulatory function as an oncogene in ESCC (*Sun et al., 2018*). These results suggest that LTA4H may bind lncRNAs to participate in transcriptional and post-transcriptional regulation to promote cancer development and result in a poor prognosis. It could reveal a novel mechanism by which LTA4H regulates LSCC and may become a possible target for clinical treatment of laryngeal carcinoma. If further studies in LSCC cells and clinical samples clarify the regulatory function of LTA4H and lncRNA interaction, new ideas will be provided for clinical treatment. In addition, we plan to conduct functional studies of LTA4H in the future, such as comparing laryngeal cancer cells with cells that overexpress LTA4H/LTA4H knockdown, to validate the findings from the RIP-seq analyses.

## CONCLUSION

In summary, this is the first time we found that LTA4H preferentially binds to the mRNAs and IncRNAs of cancer-related functional pathway genes in tumor cells by iRIP-Seq experiments and shows enriched binding in specific intron and CDS regions of these genes. Therefore, we speculated that LTA4H not only participates in the regulation of cancer through the inflammatory mediator pathway, but also influences the production of different subtypes of proteins by binding the RNA of target genes to regulate the alternative splicing process, thus regulating the proliferation, migration, and invasion of LSCC. To better understand how LTA4H regulates the alternative splicing of target genes, more research should be done. Therefore, our discoveries could shed light on the underlying mechanism of LTA4H in promoting tumor development and provide a new potential anti-tumor target.

## ACKNOWLEDGEMENTS

Thanks to ABLife Inc. for providing the RNA-seq and iRIP-seq library creation and sequencing services.

### Funding

This work was supported by the National Natural Science Foundation of China (NSFC, No. 82060497). The funders had a role in study design, data collection and analysis, decision to publish, or preparation of the manuscript.

### Grant Disclosures

The following grant information was disclosed by the authors:
National Natural Science Foundation of China: NSFC, No. 82060497.

### Competing Interests

The authors declare there are no competing interests.

### Author Contributions

- Tianjiao Ren conceived and designed the experiments, performed the experiments, analyzed the data, prepared figures and/or tables, authored or reviewed drafts of the article, and approved the final draft.
- Song Wang conceived and designed the experiments, authored or reviewed drafts of the article, and approved the final draft.
- Bo Zhang performed the experiments, authored or reviewed drafts of the article, and approved the final draft.
- Wei Zhou performed the experiments, authored or reviewed drafts of the article, and approved the final draft.
- Cansi Wang analyzed the data, authored or reviewed drafts of the article, and approved the final draft.

- Xiaorui Zhao analyzed the data, authored or reviewed drafts of the article, and approved the final draft.
- Juan Feng conceived and designed the experiments, performed the experiments, analyzed the data, prepared figures and/or tables, authored or reviewed drafts of the article, and approved the final draft.

## Data Availability

The raw data is available in the Supplemental File and the sequences are available at NCBI GEO: GSE206470.

## Supplemental Information

Supplemental information for this article can be found online at http://dx.doi.org/10.7717/peerj.14875#supplemental-information.

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
