# Peer review of "LTA4H extensively associates with mRNAs and lncRNAs indicative of its novel regulatory targets"

_PeerJ, doi:10.7717/peerj.14875_

## Round 0.1 · original submission · Major Revisions

Please address the concerns of both reviewers and revise the manuscript accordingly.

Reviewer 1 ·

Basic reporting

no comment

Experimental design

In this study, the authors report on the RIP-seq data on the protein LTA4H in HeLa cells. While the findings may be of certain merit, overall the study design has several caveats that limit its value.

1. This study uses HeLa cells, which are of cervical origin. The suitability of this cell line in this study is questionable since the paper mainly talks about LSCC.
2. The authors start the results section by showing that LTA4H level is decreased in LSCC, which is unexpected. However, instead of studying the effect of LTA4H downregulation, the authors go on to use a system where LTA4H is overexpressed to study its RNA binding. The rationale behind this is unclear.
3. The expression level of LTA4H overexpression is unknown, and its relevance to disease modeling therefore is not clear. Highly expressed proteins can cause non-specific binding to RNAs. A study using an antibody targeting endogenous proteins or endogenous tagging of the protein may provide better insights into the function of the protein.
4. Most importantly, the study lack functional studies to validate the findings from the RIP-seq analyses. For example, mutating the binding motifs of the candidate RNA is needed to validate the interactions. Comparison of wt cells vs cells with overexpressed LTA4H/LTA4H knockdown will also be informative on the function of LTA4H.

Validity of the findings

no comment

Additional comments

no comment

Reviewer 2 ·

Basic reporting

The manuscript by Ren et al investigated the RNAs that are bound by LTA4H in the Hela cell line. The authors performed iRIP-seq and revealed important targets that were bound by LTA4H, including lncRNAs. The introduction and background behind the study is well researched and structured with latest studies, effectively pointing out to aim of the study. However, the citation insertions in the text should be properly placed and following the rule of including a space between the text and the citation.

Experimental design

The experimental design of the study is sound and well-executed. There are a few suggestions to improve the overall experimental designs, which I have provided in the additional comments section. The materials and methods are lucidly explained. However, the authors should provide information regarding the deposition of the raw sequencing data.

Validity of the findings

The authors have provided appropriate conclusions for all the figures. However, there is need of more data/clarification in some cases, which I have listed in the additional comments section.

Additional comments

1. The authors spend a lot of the introduction on the Laryngeal cancer biology and examined the relationship between LTA4H expression and LSCC prognosis. However, the later experiments were performed using Hela cell line, which is a breast cancer cell line. The authors need to address the disconnection here.

2. In figure 2A, the authors did correlation analysis between input and IP samples. I think it would be more meaningful if the authors could do some correlation analysis between the two IP replicates.

3. In figure 3, the authors used DESeq analysis to identify LTA4H DEGs, which is inappropriate in my opinion. DESeq is normally used to compare RNA expression profiles between different genotypes or tissues to identify differentially expressed genes. In the case of iRIP-Seq data analysis, peak-calling algorithms are the right one to use. And the authors have already used ABLIRC algorithm in figure 2.

4. In figure 4A, the IP_2 samples showed n.s. in qRT-PCR. It would be beneficial if the authors could comment or note on this a bit.

5. The authors need to add citations for the iRIP-Seq methodology and the ABLIRC algorithm.

6. Typo at line 239, “overall” should be “overall”.

---

## Round 0.2 · accepted · Accept

The original Academic Editor is no longer available so I have assessed the revision in my role as Section Editor. Both reviewers are satisfied with the revision, therefore amended version is acceptable now.

Reviewer 1 ·

Basic reporting

The authors have successfully addressed my previous comments. Therefore, I would recommend publication of this work.

Experimental design

No comment.

Validity of the findings

No comment.

Additional comments

No comment.

Reviewer 2 ·

Basic reporting

All good

Experimental design

Good

Validity of the findings

Good